

# Why is nonword reading so variable in adult skilled readers?

Max Coltheart[1] and Anastasia Ulicheva[2]

[1] ARC Centre of Excellence in Cognition and its Disorders (CCD), and Department of Cognitive Science, Macquarie University, Sydney, NSW, Australia
[2] Department of Psychology, Royal Holloway University of London, Egham, UK

## ABSTRACT

When the task is reading nonwords aloud, skilled adult readers are very variable in the responses they produce: a nonword can evoke as many as 24 different responses in a group of such readers. Why is nonword reading so variable? We analysed a large database of reading responses to nonwords, which documented that two factors contribute to this variability. The first factor is variability in graphemic parsing (the parsing of a letter string into its constituent graphemes): the same nonword can be graphemically parsed in different ways by different readers. The second factor is phoneme assignment: even when all subjects produce the same graphemic parsing of a nonword, they vary in what phonemes they assign to the resulting set of graphemes. We consider the implications of these results for the computational modelling of reading, for the assessment of impairments of nonword reading, and for the study of reading aloud in other alphabetically written languages and in nonalphabetic writing systems.

## INTRODUCTION

Amongst the reading-related abilities possessed by adult skilled readers of alphabetically written languages is the ability to read aloud a letter string which the reader has never encountered before: a pronounceable nonword, for example. Nonword reading is of importance for theories of reading for a number of reasons.

One reason is that the ability to read nonwords—to 'sound out'—is widely regarded as very important for learning to read. Young children who are in the process of learning to read will already have an auditory vocabulary of 10,000 words or more (*Shipley & McAfee, 2015*; cited by *Law et al., 2017*) but only a very small sight vocabulary. So it will be a frequent occurrence for such children that a word they are looking at in print will not be recognizable (because it has never been seen before) whereas it would be recognizable if it were heard (because it has often been heard before). If the child were capable of print-to-sound translation for letter strings never seen before—that is, capable of nonword reading—then applying such translation would be a means by which children could capitalise on their extensive receptive spoken-word vocabularies to figure out and learn visually unfamiliar words. This is the basis of the self-teaching hypothesis about learning to read (*Share, 1995*). It is also the rationale for including phonics instruction as part of the

Corresponding author
Max Coltheart,
max.coltheart@mq.edu.au

teaching of reading, because a child without phonics ability would not be able to translate visually unfamiliar words into their familiar phonological representations, so that self-teaching could not occur.

Another reason for the theoretical importance of nonword reading is that this ability responds selectively to brain damage suffered by previously literate people. In some such people, nonword reading is selectively impaired relative to word reading: this is 'phonological dyslexia' (see *Beauvois & Dérouesné, 1979*; *Coltheart, 1996*). In other such people, the reverse is seen: nonword reading is selectively preserved relative to word reading—this is 'surface dyslexia' (see *Marshall & Newcombe, 1973*; *Patterson, Marshall & Coltheart, 1985*). The existence of this double dissociation involving nonword reading exerts powerful constraints on theorizing about reading. It is also relevant to our understanding of how children learn to read, because both phonological dyslexia and surface dyslexia exist as forms of developmental dyslexia (see *Friedmann & Coltheart, 2015*). Such findings imply that learning to read nonwords ('sounding out') is at least partly isolable from other aspects of learning to read.

Any theory of reading therefore needs to include an account of how nonword reading is accomplished. This is particularly true of computational models of reading, which originally had great difficulty in simulating nonword reading (see *Plaut et al., 1996*, p. 57).

Any attempt to account for how nonword reading is accomplished will need to confront the fact that nonword reading is very variable in adult skilled readers. For example, almost all of the 412 nonwords used in the nonword reading-aloud study by *Pritchard et al. (2012)* generated different responses from different readers. Across these nonwords, the number of different reading-aloud responses to a nonword ranged from one to 24, as shown in Fig. 1. We should emphasise here that this variability is unlikely to be associated with any difficulties in production, for two reasons. Firstly, every nonword used in this study was phonotactically and orthographically legal. Secondly, speeded responding was not required; subjects were allowed up to 10 s to make each response.

This response variability in responses to nonwords in a reading-aloud task is not a new finding.

*Masterson (1985)* gave 120 nonwords to 14 adult skilled readers to read aloud, without time pressure. Across these nonwords, the number of responses ranged from one to 10, as shown in Fig. 2.

*Calfee, Venezky & Chapman (1969)* and *Kay & Lesser (1985)* also documented this kind of variability of responses in a nonword reading-aloud task with undergraduate students. So did *Seidenberg et al. (1994)*, whose 44 undergraduate subjects, tested with 590 nonwords, produced one pronunciation to 34.7% of items, two pronunciations to 45.9%, three pronunciations to 16.9%, and four or more pronunciations to 2.5%. Similar results were reported by *Andrews & Scarratt (1998)*, whose 24 undergraduate subjects produced from one to seven different pronunciations for 216 nonwords. More recently, *Mousikou et al. (2017)* administered 915 disyllabic nonwords to 41 undergraduate subjects for reading aloud, and for each pair of subjects calculated the percentage of nonwords for which the two subjects produced the same response. Across all possible

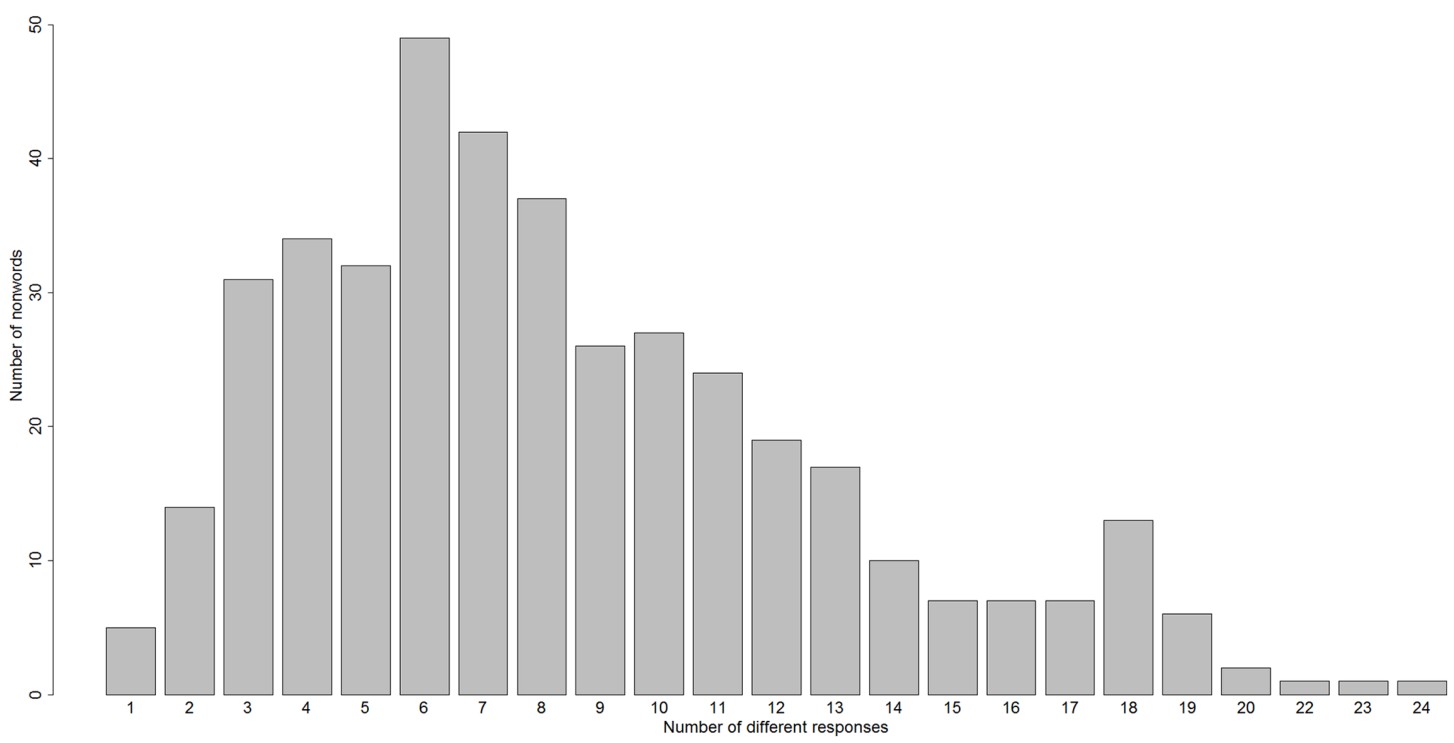

**Figure 1 A histogram of the number of different reading-aloud responses given to the 412 nonwords of _Pritchard et al. (2012)_.**

subject pairs this proportion varied from just over 40% to just under 70% (see _Mousikou et al., 2017_; Fig. 5), indicating very substantial disagreement across subjects in how nonwords should be read aloud.

How might such variability in nonword reading be explained? This what we sought to investigate.

There are various possibilities. One relatively uninteresting possibility is that the variability is largely unsystematic. Since there is no objective criterion for classifying a reading-aloud response to a nonword as correct or incorrect, it might be that what subjects do when attempting to perform this task is largely unconstrained and noisy, and little that is systematic will be discovered if such responses are scrutinised. A view of this kind has been proposed by _Forster (1985_, p. 711): 'I'm not at all sure that nonwords "have" a pronunciation. That is, I believe that asking what is the pronunciation of _tik_ is about as sensible as asking what its meaning is. What you are saying is this: If _tik_ were an English word, how would it be pronounced? Since this is a counterfactual, it is logically no different from the question: If _tik_ were an English word, what meaning would it have? Obviously a much better guess could be made about "the" pronunciation than the meaning. But it is just a guess nevertheless. Letter sequences "have" a pronunciation only if they spell words.' Here Forster, perhaps at least partly with tongue in cheek, is challenging the common assessment practice of considering that a nonword has only one correct reading-aloud response (a point we return to in the 'Discussion') by proposing that a nonword has **no** correct reading-aloud response.
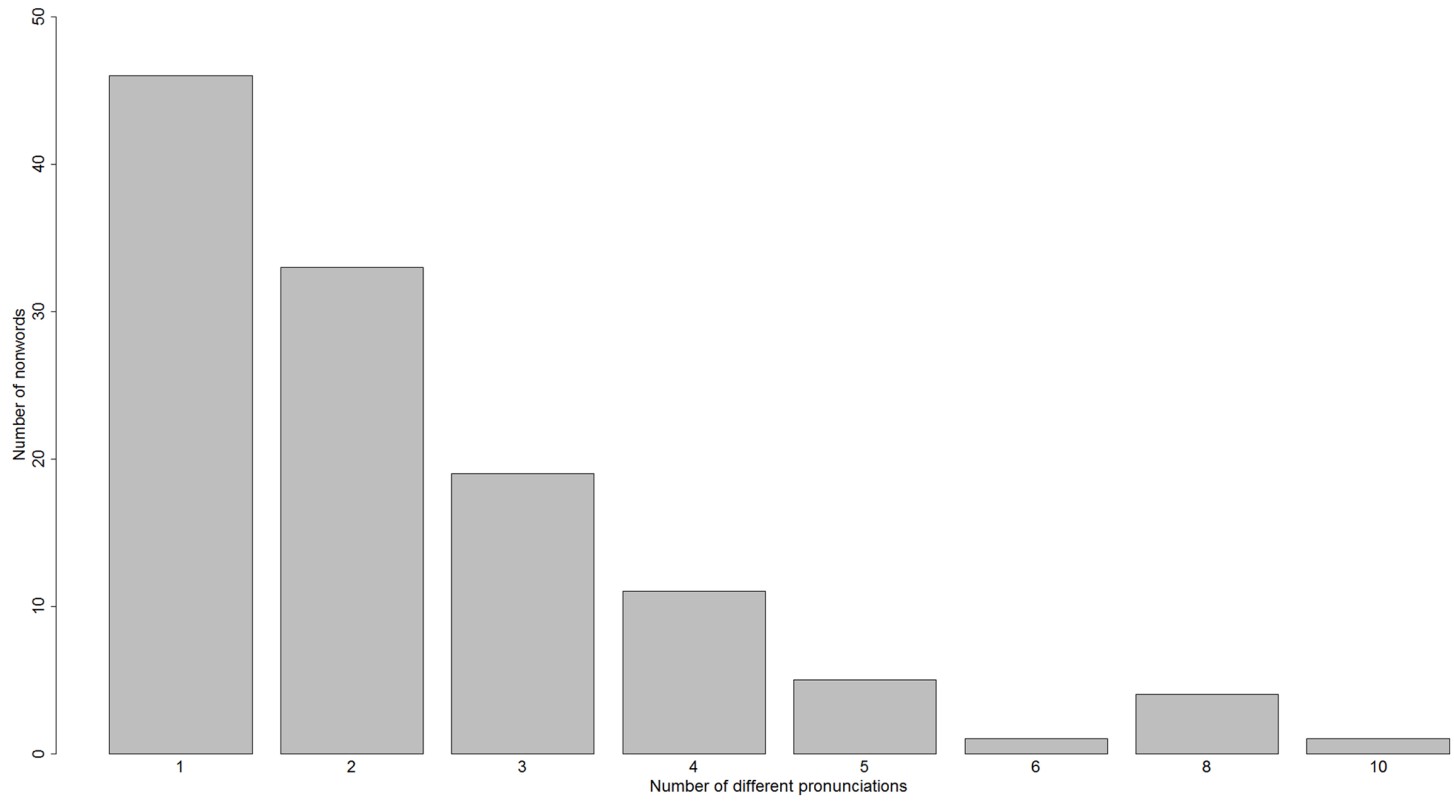

**Figure 2  A histogram of the number of different reading-aloud responses given to the 120 nonwords of *Masterson (1985)*.**

A different possibility is that there *is* a systematicity in the way people read nonwords. If there were such systematicity, then what is it that varies systematically? *Coltheart (1987*; and see Fig. 3*)* proposed that the procedure that is involved in reading a nonword like SHOIL aloud can be considered to involve at least three components. First there is graphemic parsing (producing the three graphemes SH, OI, L), followed by phoneme assignment (producing the three phonemes /ʃ/, /ɔɪ/, /l/), and then finally phoneme blending to produce a unified syllable.

It appears that these three processing components are distinct because each can be selectively impaired in patients with acquired dyslexia (*Coltheart, 1987*), as follows:

The acquired dyslexic patient MS (*Newcombe & Marshall, 1985*) had a specific impairment of the graphemic parsing process: he was especially poor at reading words containing multiletter graphemes, whose single letters he frequently treated as graphemes and so assigned phonemes to them, thus producing such reading-aloud errors as FIGHT /fighʌt/, WHOM /wəhɒm/, and ADVICE /ædvɪki/ (for other such examples of MS's failures of graphemic parsing, see *Newcombe & Marshall, 1985*, Table 2.4). The same phenomenon was observed in his single-word reading comprehension tasks (ALE -> 'kind of a path;' BARE -> 'It's an island . . . Barry Island;' SALE -> 'name of a woman, I used to fancy her, Sally').

The acquired dyslexic patient WB (*Funnell, 1983*) had a specific impairment of the phoneme assignment process. He was unable to read aloud any nonwords, and in

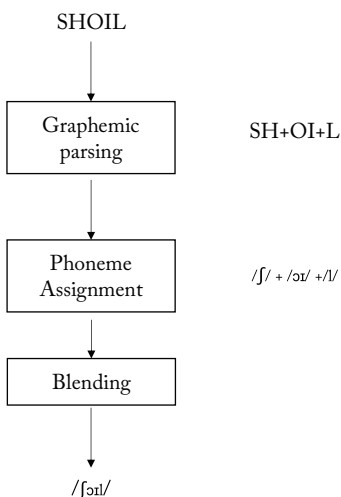

**Figure 3 A model of a grapheme-to-phoneme conversion system for reading aloud (adapted from *Coltheart, 1987*, Figure 1.3, p. 16).**

particular was completely unable to assign phonemes to single letters, even though his letter *naming* was almost perfect.

The acquired dyslexic patient CB (*Coltheart, 1987*) had a specific impairment of the blending process. Her word reading was good (93% correct), but her nonword reading was very bad (6% correct) and many of her attempts at reading three-letter nonwords involved separately pronouncing the individual phonemes associated with the graphemes of a nonword but then failing to blend the resulting set of phonemes into an integrated syllable (e.g. VOT '/və/ ... /əʊ/ ... /vəʊ/ ... /vəʊt/ ... /və/ ... /əʊ/ ... /tə/').

The relevance to our paper of these results from acquired dyslexia is that they support the model shown in Fig. 3, and our analyses in this paper are based upon that model, because we are reasoning that skilled adult readers may differ systematically in their ability to execute each of the three stages of nonword reading depicted in Fig. 3, and hence that individual differences at each of these stages could contribute to the variability of nonword reading responses.

We might expect three subject-based variables to be relevant here:

(1) Differences in graphemic parsing (do subjects differ in how they parse a particular nonword?).

(2) Differences in phoneme assignment (do subjects differ with respect to what pronunciations they assign to particular graphemes?). This is the possibility that *Seidenberg et al. (1994*, p. 1179*)* were referring to when they said: 'The fact that different pronunciations are generated across subjects can be explained by assuming that they have slightly different rule sets.'

(3) Differences in blending (do subjects differ with respect to how they blend the phonemes of a nonword?).

The three subject-specific variables mentioned above might explain some of the variability on response made when nonwords are being read aloud, but there might also be

item-specific variables influencing this variability. In Figs. 1 and 2, one can clearly see that some items are read unanimously by all subjects, while other items evoke much variability. So, just as there might be variability across subjects in how variably they read nonwords, there appears also to be variability across nonwords in how variably they are read. If we take such an item-based perspective, three item-based variables that might be relevant here are:

(4) Differences in graphemic parsing (do nonwords differ with respect to how variably they are parsed?).
(5) Differences in phoneme assignment (do nonwords differ with respect to how variably phonemes are assigned to their graphemes?).
(6) Differences in blending (do nonwords differ with respect to how variably they are blended?).

Our goal in this paper is to explore the degree to which variability in reading-aloud responses to nonwords is influenced by each variable.

## METHOD

Our analyses are based on existing data that were collected in the study by *Pritchard et al. (2012)*, referred to above. Therefore, we will first briefly summarise the details of that study.

*Pritchard et al. (2012)* began with a detailed investigation of nonword reading by two computational models of reading, the Dual-Route-Cascaded (DRC model (*Coltheart et al., 2001*) and the Connectionist Dual-Process (CDP+) model (*Perry, Ziegler & Zorzi, 2007*). This was done by obtaining each model's response to 1,475 monosyllabic nonwords chosen from the ARC Nonword Database (*Rastle, Harrington & Coltheart, 2002*). All the chosen nonwords were phonologically and orthographically legal nonwords of English in the sense that (a) none contained a phoneme sequence that is phonotactically illegal in English or occurs in only a very few English words and (b) none contained any bigram letter sequences that do not occur in any English words.

There were 412 of these 1,475 phonologically and orthographically legal nonwords for which the two models produced different responses. These are the informative nonwords here, since for each of them one can determine which (if either) model response corresponds to the response adult skilled readers make. That allows an assessment of which model's nonword reading procedure most resembles the nonword reading procedure used by adult skilled readers. Hence these 412 nonwords were given to 45 undergraduate university students to read aloud. The nonwords were presented in uppercase. There was no time pressure to respond. Each response was phonetically transcribed by two judges, one of them a trained phonetician. The set of 412 nonwords, and reading responses to them, is available at https://www.researchgate.net/publication/257938878_111020NonwordReading.

We pursued our aims by analysing the responses of the subjects in this nonword reading study. Out of 18,540 potential responses (45 subjects by 412 nonwords) in the database from this study, 422 (2.3%) were unavailable for analysis (because, for example, the

subject did not respond, or because the response could not be analysed due e.g. to unintelligibility). Hence we had 18,118 responses to analyse.

## RESULTS

### The analysis of graphemic parsing

[1] A grapheme is the written representation of a phoneme. So a word with three phonemes must have three graphemes, regardless of how many letters it has (cf. Fig. 3).

By the definition of 'grapheme'[1], the number of graphemes into which a nonword is parsed should equal the number of phonemes in that nonword's pronunciation. Given this, one would expect subjects to produce parsings where the number of graphemes in the stimulus equals the number of phonemes in the response. However, subjects might not always do this. If sometimes they do not, then different subjects would be producing different parsings, and therefore different reading-aloud responses, with the same nonword. We refer to parsings which violate the constraint that number of graphemes is equal to number of phonemes as 'nonstandard parsings.'

Examples of such nonstandard parsings from our dataset using the example nonword SPRAUK, whose standard parsing is to the graphemes <S> <P> <R> <K>, are:

(a) Response /sprʌŋk/ indicating a parsing into the graphemes <S> <P> <R> <U> <N> <K>: here a new grapheme N has been inserted.
(b) Response /spɔːk/ indicating a parsing into the graphemes <S> <P> <K>: here an existing grapheme R has been omitted.
(c) Response /sprau/ indicating a parsing into the graphemes <S> <P> <R> <AU>: here an existing grapheme K has been omitted

The number of nonstandard parsings in the set of 18,118 analysable responses was 2,199 (12.1%). Thus nonstandard graphemic parsing is not an uncommon occurrence when nonwords are read aloud.

### The analysis of phoneme assignment

For the 15,919 reading-aloud responses which elicited standard parsings (i.e. where the number of phonemes in the response was equal to the number of graphemes in the stimulus), each grapheme in the stimulus is unambiguously associated with one phoneme in the response, and vice versa. Such responses can be analysed to determine the extent to which a given subject always assigns the same phoneme to a given grapheme. They can also be analysed to determine the extent to which in any given nonword all subjects assign the same phonemes to that nonword's graphemes.

We only considered graphemes that occur in every subject's set of responses at least four times, so as to make sure a grapheme has a reasonable chance of being read in different ways by different subjects. There were 36 such graphemes: GE, A, AU, B, C, CH, D, E, E.E, F, G, H, I, I.E, K, L, LL, M, N, O, O. E, OO, OW, O, P, PH, R, S, SH, T, TH, U, U.E, V, W, Y, Z (note that context is taken into account for this analysis, that is, G and G (before E) are different graphemes). The maximum frequency of occurrence of a grapheme in a subject's set of responses was 114 (this was the grapheme L in many subjects' sets of responses).

On average, a grapheme from our target set of 36 graphemes occurred 30 times in subjects' responses (from four to 114 times). If each subject produced standard parsings for all nonwords containing any one of the 36 graphemes, we would have 1,216 analysable Grapheme-Phoneme Correspondences (GPCs) per subject (i.e., the sum of maximum across subjects frequencies of 36 graphemes). This would render 54,720 data points in total as there are 45 subjects. However, not every subject parsed every one of the 36 graphemes out on every occasion (e.g., Subject 29 parsed grapheme L out on 69/114 occasions). This left us with 48,269 grapheme–phoneme assignments to analyse.

We define 'standard phoneme assignment' for any grapheme as the phoneme that occurs most often for that grapheme in the monosyllabic words of English, taking context and position into account. Subjects did not always use standard phoneme assignments. For example, the most common phoneme for the grapheme AU is /ɔː/, but other phonemes (e.g. /aʊ/, /ʌ/, /ɑː/) were also assigned to this grapheme.

Of the 48,269 grapheme–phoneme assignments we analysed, 4,230 (8.8%) were nonstandard assignments. Thus nonstandard phoneme assignment is not an uncommon occurrence when nonwords are read aloud.

## The analysis of blending

We mentioned above that differences at the blending stage might contribute to individual variability in response to nonword. However, we did not observe this. No subject ever failed to blend individual phonemes into an integrated syllable when reading aloud a nonword. Hence, variables 3 and 6 are not contributing to variability in nonword reading, and so we will not consider these variables further.

## Subject-based variability
### Graphemic parsing variability as a contributor to nonword reading variability

Does the incidence of nonstandard graphemic parsings vary from subject to subject? The answer is Yes: Fig. 4 shows the percentage of nonstandard parsings for each subject, which varied across subjects from 3.16% (Subject 5) to 36.65% (Subject 29).

Hence, we have established that one variable that contributes to variability in nonword reading is a difference between subjects in graphemic parsing (variable 1).

### Phoneme assignment variability as a contributor to nonword reading variability

Does the incidence of nonstandard phoneme assignments vary from subject to subject? The answer is Yes: Fig. 5 shows the percentage of nonstandard assignments for each subject, which varies from 9% (Subject 1) to 31% (Subject 9).

Hence, we have established that another variable that contributes to variability in nonword reading is a difference between subjects in phoneme assignment (variable 2).

Another variable that can be considered here is *how many different* phonemes a particular subject assigns to a particular grapheme. We assessed this kind of variability by measuring entropy (H).
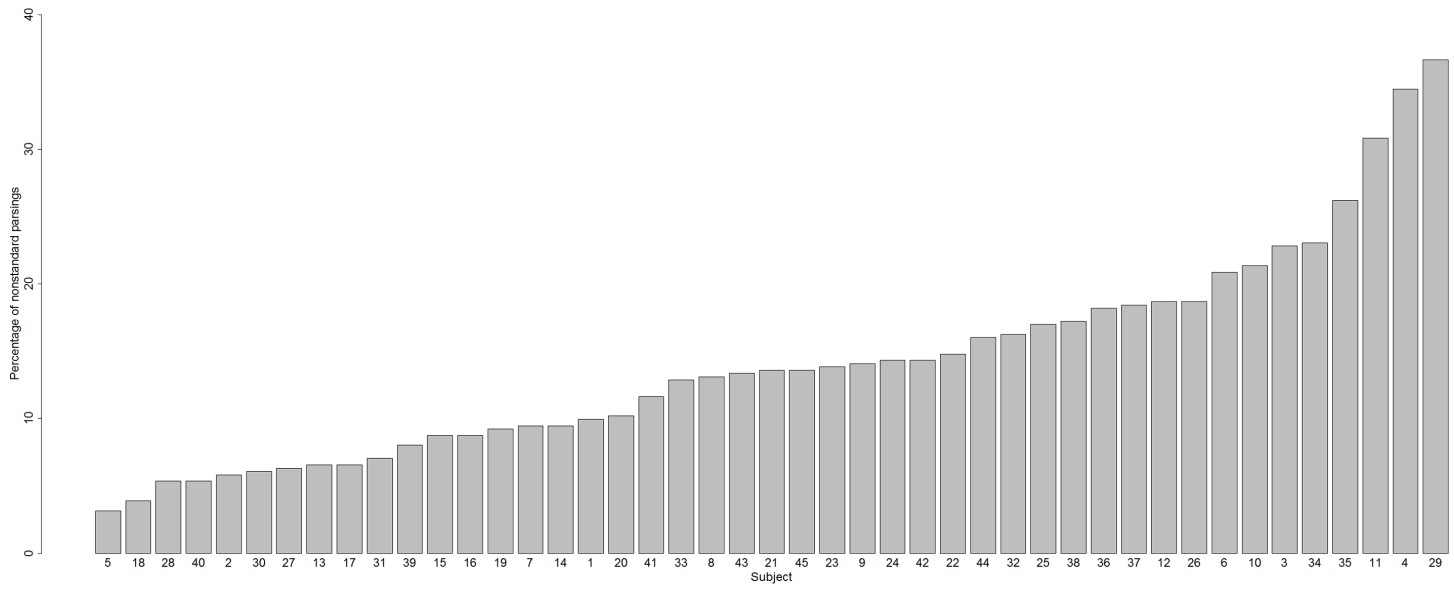

**Figure 4 Percentages of nonstandard parsings for each subject.**

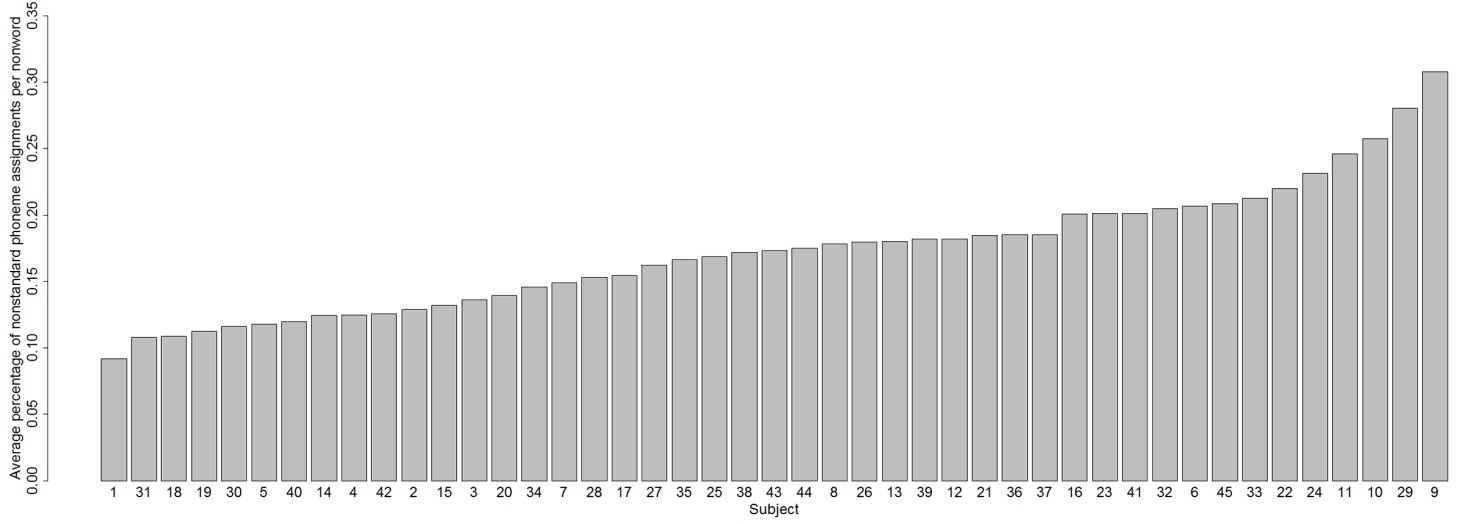

**Figure 5 Proportion of nonstandard phoneme assignments for each subject.**

To measure how variable the pronunciations given to graphemes are across all subjects, for each grapheme we calculated entropy H using the following formula from *Zevin & Seidenberg (2006)*:

$$H = \sum [-p_i x \log_2(p_i)]$$

where $p_i$ is the proportion of participants assigning the grapheme a particular phoneme. An H value of 0 denotes that participants were unanimous when assigning a phoneme to a particular grapheme, whereas high H values indicate high variability across subjects in what phoneme was assigned to that grapheme.

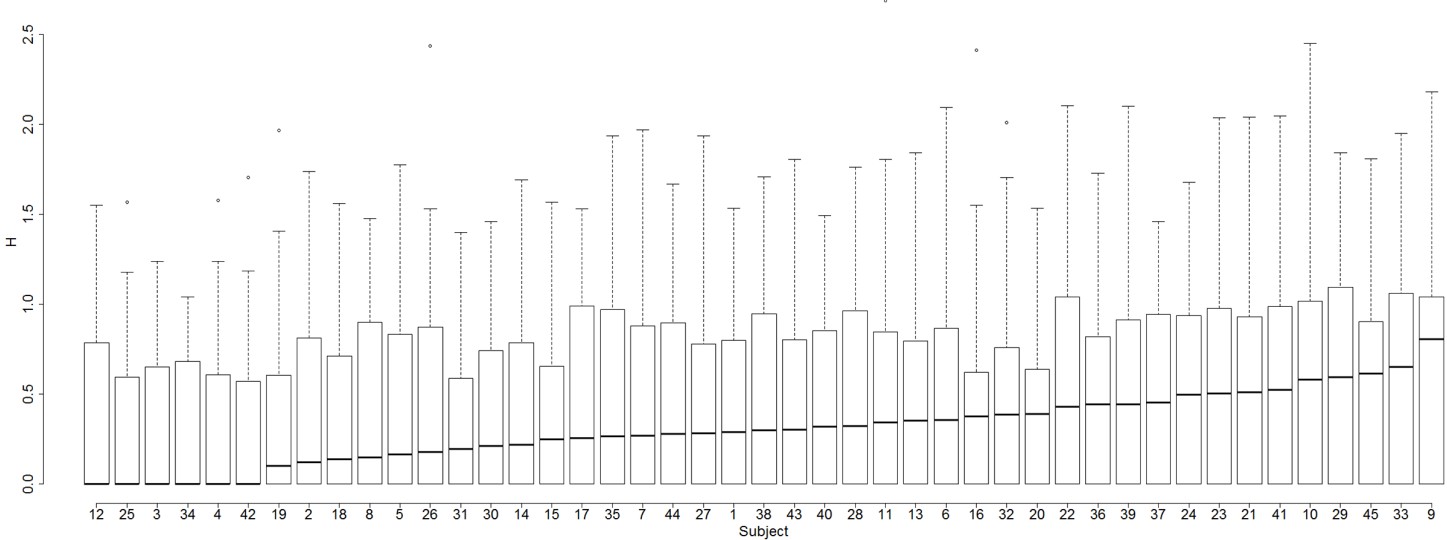

**Figure 6 Box-and-whisker plots demonstrating variability across subjects in entropy of assignment of phonemes to graphemes.**

The value of H for each the 36 graphemes was calculated for each of the 45 subjects. Does phoneme assignment variability as indexed by H differ from subject to subject? Figure 6 indicates that the answer is Yes.

Every subject had an H value of 0 for at least one grapheme: so for every subject there was at least one grapheme to which that subject consistently assigned the same phoneme. But as Fig. 6 shows, the degree to which such consistency is seen varies considerably across subjects: some subjects have a number of graphemes for which H is very high and some do not.

For example, Subject 42 had a mean H of 0.25—this subject tends not to assign many different phonemes to the same grapheme. In contrast, Subject 9 had a mean H of 0.71—this subject produces many different phonemes in response to some graphemes. To illustrate, both subjects always assigned the same phonemes to the graphemes B, D, F, Z in a consistent way, but whereas Subject 42 was also consistent at assigning phonemes to the graphemes GE, C, G, L, O, P, PH, R, T, CH, I.E, Subject 9 exhibited much variability for each of these graphemes—the last two graphemes were particularly unstable, with H values of 1.68 and 1.84, respectively (for these graphemes Subject 42 had Hs of 0). Subject 9 translated CH as /tʃ/ (three times), /k/ (11 times), /s/ (once), /ʃ/ (twice), /θ/ (once) (for Subject 42 CH is always /tʃ/), and I.E as /aɪ/ (once), /ɛ/ (once), /i/ (twice), /ɪ/ (three times) (for Subject 42 I.E was always /aɪ/).

Hence, we have established not only that subjects vary greatly in the degree to which they assign the standard phoneme to a grapheme, but also that they vary greatly in how many different phonemes they assign to a given grapheme (i.e., they vary in the entropy of phoneme assignments).
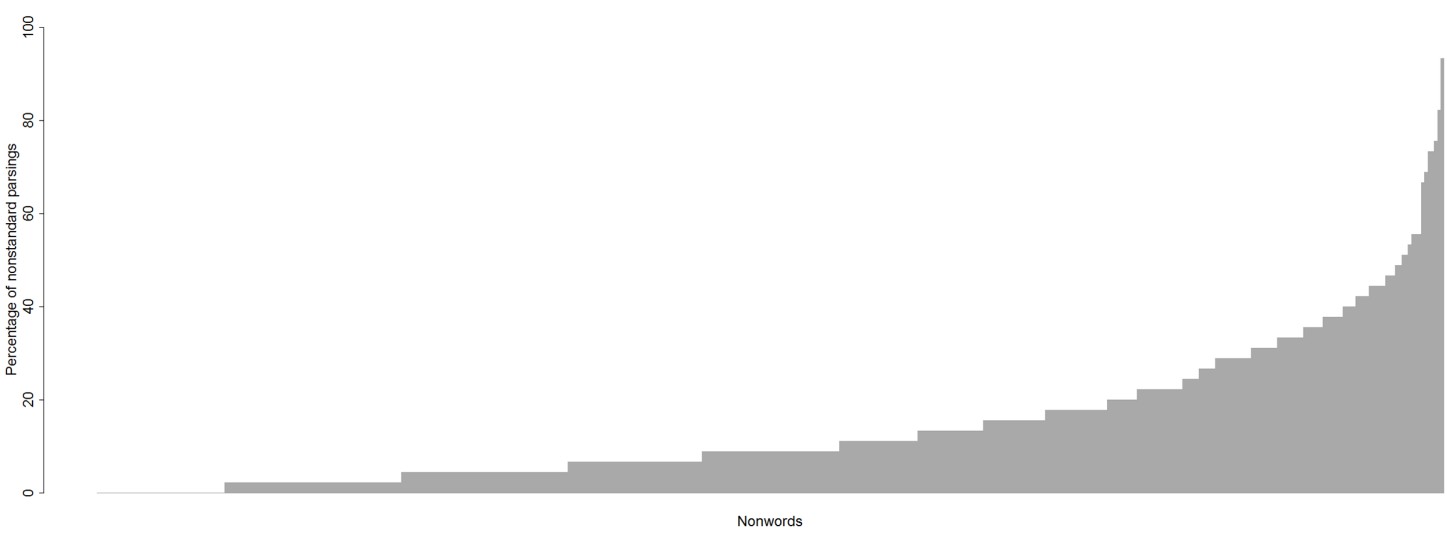

**Figure 7** Percentage of nonstandard parsings for each nonword.     

## Item-based variability

In this section we discuss variability in graphemic parsing and phoneme assignment from an item-based perspective.

### Graphemic parsing variability as a contributor to nonword reading variability

Does the way people perform graphemic parsing different from nonword to nonword? Yes. The percentage of nonstandard parsings varies across nonwords from 0% to 93.33% (see Fig. 7).

Thirty-nine nonwords out of 412 were always parsed in the standard way, by all participants. These included nonwords with single-letter graphemes such as BREC, MOLF, and NOF, but also nonwords with multi-letter graphemes such as OOSH, SNOWL, and THUSE.

Nonwords like DONGE (graphemes D, O, N, GE), SCROME (graphemes S, C, R, O.E, M), and GANC (graphemes G, A, N, C) evoked more variability, and produced the median level (9%) of nonstandard parsings (e.g. three-phoneme /dɒŋ/ instead of the standard four-phoneme /dɒndʒ/, /skrə/ instead of /skrəʊm/, /gæŋ/ instead of /gæŋk/, respectively).

Nonwords with more than 50% of nonstandard parsings were, for example, GNEUTH (standard parsing is into the graphemes GN, EU, TH yielding the pronunciation /nuθ/) which had nonstandard parsings as in /gwɛnθ/, /knut/; PSIRP (standard parsing is into the graphemes PS, IR, P yielding the pronunciation /sɜːp/) which had nonstandard parsings as in /psɜːp/, /psɪrəp/; and CLALF (standard parsing is into graphemes C, L, A, L, F yielding the pronuncaition /klælf/) which had the nonstandard parsing as in /klæf/.

A prominent feature of the nonstandard parsing data was that a multiletter vowel grapheme was often assigned two or more phonemes rather than one because the

**Table 1 Examples of responses in which a multiletter vowel grapheme was parsed into more than one grapheme each of which was assigned a phoneme.**

| Nonword | Example word | Multiletter vowel grapheme | Number of letters in grapheme | Response indicating nonstandard parsing of grapheme |
|---|---|---|---|---|
| Psoath | Oath | oa | 2 | səʊwəθ |
| Gluit | Fruit | ui | 2 | gluwit |
| Gwene | Scene | e.e | 2 | gwini |
| Twole | Stole | o.e | 2 | twɒlɪ |
| Trure | Pure | u.e | 2 | trurɪ |
| Thaque | Plaque | a.ue | 3 | θækjuː |
| Waice | Plaice | ai.e | 3 | waɪtʃɪ |
| Hauve | Mauve | au.e | 3 | haʊvɪ |
| Strique | Clique | i.ue | 3 | strikɪ |
| Hiece | Piece | ie.e | 3 | haɪtʃɪ |
| Wouge | Rouge | ou.e | 3 | wuʤɪ |
| Crusque | Brusque | u.ue | 3 | kruzkjuː |
| Frugue | Fugue | u.ue | 3 | frugjuː |
| Pseuce | Deuce | eu.e | 3 | sutʃɪ |
| Suile | Guile | ui.e | 3 | suwəl |
| Stoarse | Coarse | oa.e | 3 | stɔːwɑːs |

grapheme was parsed as not one, but two graphemes, on the basis of its constituent letters. Examples are given in Table 1.

This phenomenon was also seen with multiletter consonant graphemes as in the example *gneuth* /gənuθ/, where the single grapheme *gn* was nonstandardly treated as two graphemes. However, this occurred far less often than it did for multiletter vowel graphemes. The failures of graphemic parsing illustrated in Table 1 were seen also in the acquired dyslexic patient MS (*Newcombe & Marshall, 1985*), discussed above.

Hence, we have observed that the percentage of nonstandard parsings varies from nonword to nonword: so variable 4 does contribute to variability in nonword reading.

### Phoneme assignment variability as a contributor to nonword reading variability

Does the way people perform phoneme assignment differ from nonword to nonword? This question cannot be addressed directly, because different nonwords consist of different graphemes. Instead, we can ask—is the variability of phoneme assignment different from grapheme to grapheme? If we find that some graphemes evoke more variability in phoneme assignment (i.e., have a high entropy) across subjects than others, then it is legitimate to infer from this that nonwords containing these graphemes will also be pronounced more variably than those consisting of graphemes with a low entropy, which gives us the answer to our question.
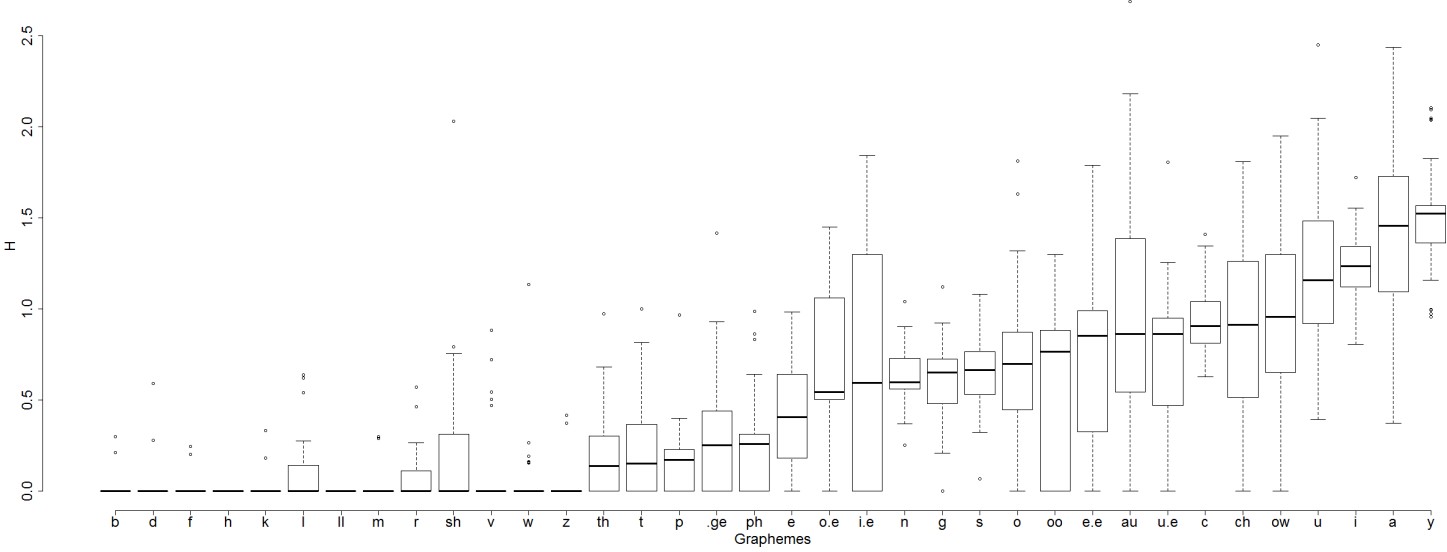

**Figure 8** Box-and-whisker plots demonstrating variability across graphemes in entropy of assignment of phonemes to graphemes.

So for each of the 36 graphemes that a subject parsed out at least four times, we calculated grapheme entropy H across subjects i.e. how variable across subjects are assignments of phonemes to particular graphemes. Does phoneme assignment entropy differ from grapheme to grapheme? Figure 8 indicates that this is so.

The mean value of H varies across graphemes, from 0 (for two of the 36 graphemes, LL and H, all subjects assigned the same phoneme to the grapheme) to 1.498 (grapheme Y). For example, the number of phonemes assigned to the grapheme I.E was highly variable across subjects. Some subjects consistently assigned just one phoneme to this grapheme in all nonwords that contained the grapheme (13 subjects always read I.E as /aɪ/). Other subjects were highly inconsistent in reading this grapheme in different nonwords (H = 1.84 in Subjects 13 and 9, e.g. Subject 9 assigned the phonemes /ɪ/, /i/, /aɪ/, and /ɛ/ to the grapheme I.E in different nonwords containing that grapheme.

This indicates that graphemes vary with respect to how variable the phonemes assigned to them are, and so grapheme entropy (variable 5) is contributing to variability in nonword reading.

## Summary of our findings

We have thus identified five factors which contribute to the variability of responding seen when skilled readers read nonwords aloud:

(a) There is a great deal of variability across subjects in their tendency to graphemically parse nonwords in the standard way;

(b) There is a great deal of variability across subjects in the probability that they will assign the standard phonemes to graphemes;

(c) There is a great deal of variability across subjects in the number of phonemes they assign to particular graphemes;

(d) There is a great deal of variability across nonwords in their tendency to yield standard graphemic parsings;

(e) There is a great deal of variability across graphemes in the number of different phonemes assigned to graphemes.

## DISCUSSION AND CONCLUSIONS

Although we would not follow *Forster (1985)* quite so far as to agree with his suggestion that any subject's attempt at deriving phonology from a printed nonword is 'just a guess,' we do agree with him that there is no such thing as ***the*** correct response when a subject is attempting to read a nonword aloud, simply because of the substantial variability on the actual responses produced when skilled readers of English read nonwords aloud.

In this paper we attempted to identify sources of the substantial variability in how people read English nonwords aloud. We adopted the account of nonword reading in English offered in *Coltheart (1987*; see Fig. 3*)*. According to this account, the procedure that is employed for the reading aloud of nonwords consists of three components: grapheme parsing, followed by phoneme assignment, followed by phoneme blending. Variability in the operation of any one of these components could contribute to the observed variability in nonword reading.

Variability in phoneme blending made no contribution here, because there was no such variability: all subjects always produce a blended set of phonemes (i.e., an integrated syllable) when reading aloud a nonword. In contrast, there was a great deal of variability, across subjects and across items, of both the graphemic parsing stage and the phoneme assignment stage. So our answer to the question 'What are the reasons for the variability in nonword reading in healthy adult readers?' is

(1) Between-subject differences in grapheme parsing, i.e., subjects vary greatly in the degree to which they produce nonstandard graphemic parsings;

(2) Between-item differences in grapheme parsing, i.e., nonwords vary greatly in the number of different graphemic parsings the nonword yielded;

(3) Between-subject differences in phoneme assignment, i.e., subjects vary greatly in the degree to which they assign the standard phoneme to each grapheme, and also in the degree to which they assign a variety of different phonemes to any particular grapheme;

(4) Between-item differences in phoneme assignment, i.e., items vary greatly in the degree to which the standard phoneme to their graphemes, and also in the degree to which a variety of different phonemes is assigned to their graphemes.

These are the factors which underlie the variability of responses in the task of reading aloud nonwords.

### Implications for the computational modelling of reading

*Pritchard et al. (2012)* found that nonword reading responses generated by a grapheme–phoneme conversion procedure (as used by the DRC model) were much more similar to human responses than responses generated by a neural network procedure trained by the

Delta Rule learning algorithm (as used in the CDP+ models). However, for 26.5% of the nonwords used, the most common human response to these nonwords was not the GPC-based response that the DRC model produces (see *Andrews & Scarratt, 1998*, for similar results). For the CDP+ model, 87.9% of the most common human responses differed from the model's responses here. Thus there is substantial variance of nonword reading that is not captured by these models.

Our analyses in this paper have shown that some of this uncaptured variance is due to variability in what phonemes are assigned to graphemes. In the present version of DRC model, only one phoneme is associated with each grapheme, and we have observed here that there are only a few graphemes to which every human reader assigns a constant phoneme. This issue might be dealt with in future modelling work by investigating the suggestions by *Seidenberg et al. (1994)* and *Zevin & Seidenberg (2006)* that different skilled readers have slightly different GPC rule sets. If this turns out to be so, then multiple versions of the DRC model could be produced, each with a slightly different GPC rule set, in an effort to claim some of the currently uncaptured variance in nonword reading data.

Our analyses in this paper have also shown that some of this uncaptured variance is due to variability in the process of graphemic parsing. In the present version of the DRC model, there is only one way to parse each letter string. Analogous to the approach suggested in the previous paragraph, we can consider the possibility that different skilled readers have slightly different *grapheme* sets. Perhaps subjects who read aloud the nonword *gwene* as '/gwini/' (see Table 1 for this and other comparable examples) do not have the grapheme E.E in their set of graphemes, and so treat these two letters as two graphemes rather than as a single grapheme.

## Implications for the assessment of acquired and developmental phonological dyslexia

Acquired phonological dyslexia was first described by *Beauvois & Dérouesné (1979)* and developmental phonological dyslexia first described by *Temple & Marshall (1983)*. In this and subsequent work, this condition was normally diagnosed on the basis of the reading aloud of nonwords being less accurate than the reading aloud of words (see *Berndt et al., 1996*, Table 2 and Appendix). But if we take the view that the correct reading-aloud response for a nonword cannot be defined, then the accuracy with which a group of nonwords is read aloud also cannot be defined. How, then, can we decide whether a person with poor reading should be classified as exhibiting phonological dyslexia?

A new approach to identifying acquired or developmental phonological dyslexia is therefore needed. One issue that may be important here is that many of the nonwords used by *Pritchard et al. (2012)*, though all were monosyllabic and orthographically and phonologically legal, were orthographically and phonologically rather complex. In contrast, many of the nonwords used in standardised assessments of nonword reading such as those provided in the PALPA battery (*Kay, Lesser & Coltheart, 1992*), the MOTIf battery (www.motif.org.au), the Woodcock Word Attack subtest (*Woodcock, McGrew &*

*Mather, 2001*) and the Phonemic Decoding Efficiency component of the Test of Word Reading Efficiency (*Torgesen, Wagner & Rashotte, 1999*) are rather simple. It may be, then that when the nonwords from these batteries are administered to appropriate controls (skilled adult readers, or children whose learning to read is progressing normally), there would be much less variability of response. Our results suggest nevertheless that there would not be unanimity of response by all control readers for all of these nonwords. There will be nonwords which evoke different responses in different control readers. Should all such responses be scored as correct when nonword reading is being assessed?

That being said, the widespread practice of scoring the reading-aloud response to a nonword as correct only if it conforms to the standard GPCs of English has much to recommend it (even though it will lead to many responses that control readers actually do make being classified as errors) when children's reading is being assessed. This is because a critical component of reading acquisition is the child's ability to correctly derived phonology from print when the child encounters a word on the page that has never been seen before. Application of standard English GPCs will not achieve this for all words, but it will for the majority of such words (over 80% of monosyllabic words, for example), and therefore is a productive strategy that will assist learning to read. For that reason, it is important to assess just how well a child can produce GPC-governed responses when reading nonwords aloud.

## Implications for other alphabetically written languages and for nonalphabetic writing systems

The correspondences between orthography and phonology are more complex and more subject to exceptions in English than is the case for any other language that is written alphabetically. Might this be one reason for the variability of nonword reading that we have documented here? What might one see if a nonword reading study corresponding to that of *Pritchard et al. (2012)* were carried out with readers of a much more regularly spelled language such as Italian or Spanish? Would there be much greater uniformity of response?

One might expect an even great deal of uniformity in nonword reading aloud when the script used is a syllabic one such as Japanese hiragana or katakana. This is because graphemic parsing, one source of variability in nonword reading, is not needed when reading these syllabic scripts, as there is one-to-one mapping from each individual kana character to its pronunciation. Alphabetic scripts are different because in most such scripts some phonemes are represented by a *set* of letters rather than just one letter. Any such letter set has to be treated as a unit, i.e., as a grapheme). What is more, the nonlexical mapping of hiragana or katakana characters to their pronunciations is fixed in Japanese: there are no words which disobey the standard mappings; that might be expected to reduce or even eliminate variability in reading nonwords written in hiragana or katakana.

### Funding

Anastasia Ulicheva holds an ESRC Future Research Leaders Fellowship (ES/N016440/1). The funders had no role in study design, data collection and analysis, decision to publish, or preparation of the manuscript.

### Grant Disclosures

The following grant information was disclosed by the authors:
ESRC Future Research Leaders Fellowship: ES/N016440/1.

### Competing Interests

The authors declare that they have no competing interests.

### Author Contributions

- Max Coltheart authored or reviewed drafts of the paper, approved the final draft.
- Anastasia Ulicheva analysed the data, prepared figures and/or tables, authored or reviewed drafts of the paper, approved the final draft.

### Data Availability

Data from Pritchard et al. is available at https://www.researchgate.net/publication/257938878_111020NonwordReading and uploaded as a Supplemental File.

### Supplemental Information

Supplemental information for this article can be found online at http://dx.doi.org/10.7717/peerj.4879#supplemental-information.

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
