# Peer review of "Why is nonword reading so variable in adult skilled readers?"

_PeerJ, doi:10.7717/peerj.4879_

## Round 0.1 · original submission · Minor Revisions

I thank you for submitting your work to PeerJ. I have received reviews from two experts in the field, both of whom are positive about the manuscript (as am I). You will find their feedback below. I will highlight a few items from their reviews that resonated with my reading of the piece. Reviewer 1 noted that your manuscript could benefit from a section devoted to Methods. I concur with this assessment. A general overview of your dataset and analytical procedures would help to preemptively bootstrap the reader's appreciation of your Results. Reviewer 1 also noted that some of your sentences are lengthy and overly-complicated. Indeed, I found myself rereading many sentences multiple times before appreciating what was being communicated. These, of course, are relatively small issues that should be easily addressed.

Reviewer 1 took issue with your quotation from Ken Forster. I quite liked the quote and found that it provided a nice bookend to your piece. Perhaps, per the suggestion of Reviewer 1, you could further unpack the quotation so that its relevance to your analyses is clarified.

I look forward to reading a revised version of this document.

Reviewer 1 ·

Basic reporting

The abstract says that the analysis of the nonwords led to the identification of two factors that explain nonword reading variability. However, you were looking specifically at those factors so it is incorrect to say the analysis led to the identification of them. This would need revising to ensure it matches what was reported.

Experimental design

The literature review is comprehensive and informs the basis of the research itself. There were places where I was unclear on what the authors intended. On the first page of the literature review (lines 69-73) it essentially says that if a child could simply read a word they didn’t know they would be using their receptive vocabulary. This seems to be an underdeveloped point, especially as it relates to the next sentence arguing it is the basis for Share’s self-teaching hypothesis and for teaching phonics. This paragraph may need to be reworded to more clearly make the point regarding the relevance of nonword (unfamiliar word) for theories of learning to read.
The structure of the literature review does not help with clarity. There is an indepth explanation of the Pritchard data set in the middle of the literature review, which detracts from the main claim that nonword reading responses are variable. Perhaps it would be better for the explanation of the Pritchard data set to take place in the (currently not there) methodology section. This would make the literature review easier to follow. The findings of the Pritchard study should still be elaborated on as a reason to examine the human responses from the Pritchard study. However, this may need to be better integrated with the other studies that found variability. For example, there is a two sentence reference to Masterson and a figure from it, but this is not clearly explained in terms of its inclusion.
The purpose of the reference to Forster, and the relatively long quote, is not made clear. Perhaps the point could be made without the quote. The key argument appears to be that nonword reading variability can be explained by differences in graphemic parsing and phonemic assignment. I appreciate the reason for the examples of the acquired dyslexics, but I would suggest that there be a better link between that and normal readers. Otherwise, the reference to acquired dyslexia and the normal adult readers in the database used, is vague.
The subject-based variables explained in lines 231-241 do follow on from the previous discussion. However, the description of the three item-based variables (lines 240-254) is not as well explained in the prior section. This may need to be better related to the preceding literature review.
There are some complex and long sentences within the writing (e.g. lines 274-277). I found many of these sentences difficult to follow and reduced clarity in the authors’ argument.
I found the overall design of the study to be very interesting and a natural follow-on from the Pritchard et al. study.
Structurally I believe it would be helpful to have a methodology section in which the description of the data set and data analysis procedures are given would allow the results section to be more concise in the coverage of the analysis.
Why did Table 1 not have a column indicating the number of phonemes used in the phoneme assignment analysis? This would help with the interpretation of the phoneme assignment analysis.

Validity of the findings

No Comment

Additional comments

It is stated (lines 609-617) that graphemic parsing explains some of the variability in nonword reading. I note that graphemic parsing was measured by the omission and insertion of graphemes (as measured by phoneme assignment missing or added; lines281-292). In this way it is when phonemes are more or less than graphemes. There is a great deal of variability in this, particularly in terms of the difference between turning one vowel digraph grapheme into two phonemes (mentioned in lines 614-617) and adding an entirely new phoneme as in example a (lines 285-287). Perhaps the definition of non-standard grapheme parsing needs revising, for clarity.

·

Basic reporting

No comment

Experimental design

No comment

Validity of the findings

A possible improvement is to comment more on "grapheme sets". The researchers say that different adults may have different grapheme sets, e.g., whether or not they are aware of the silent e pattern as in “gwene” to help with reading of this nonword. It does seem to explain why some of the 45 undergraduates were so variable in responses. Some their grapheme sets may be due to a high level of implicit learning, that is, statistical learning of GPCs through reading of text rather than routine application of GPCs when identifying words. Thus, many of the graphemes in the test material might give difficulty even though they were legal graphemes since they might not have been seen before. It would be interesting to compare adults who did and did not learn GPC rules at school and whether they would make the same mistakes. Further studies might look at this question. It might throw light on the possible value of GPC knowledge for reading acquisition over the long term.

Additional comments

it is a nicely written study.

---

## Round 0.2 · accepted · Accept

Thanks so much for submitting a revision to the manuscript. I asked Reviewer 1 from the first round to have a look, and they were quite satisfied (as you will see below). I think this piece makes an interesting contribution to the literature. Thanks for publishing with PeerJ!

# Reviewer 1 ·

Basic reporting

I have no further comments to add.

Experimental design

I have no further comments to add.

Validity of the findings

I have no further comments to add.